# Factors Influencing the Educational Needs and Nursing Intention Regarding COVID-19 Patient Care among Undergraduate Nursing Students

**DOI:** 10.3390/ijerph192315671

**Published:** 2022-11-25

**Authors:** Eun-Joo Ji, Eun-Kyung Lee

**Affiliations:** 1Department of Nursing, Catholic Kwandong University, Gangneung 25601, Republic of Korea; 2College of Nursing, Daegu Catholic University, Daegu 42472, Republic of Korea

**Keywords:** COVID-19, needs, intention, attitude

## Abstract

Purpose: This study examines final-year undergraduate nursing students to determine the educational needs for Coronavirus disease 2019 (COVID-19), knowledge of COVID-19, attitude toward COVID-19 patient care, and nursing intention toward COVID-19 patients. Methods: A structured questionnaire was used to collect data from 21 April to 6 May 2022. The participants included 144 final-year (4th year) undergraduate nursing students in Gangwon-do, Daegu-si, and Chungcheong-do. The SPSS/WIN 21.0 program was used to analyze the data; Pearson’s correlation coefficients and multiple regression were further performed. Results: The attitude toward COVID-19 patient care (β = 0.38, *p* = 0.001), a cohabitant formerly infected with COVID-19, and the intention to study COVID-19 patient care (β = 0.16, *p* = 0.018) were found to influence nursing intention toward COVID-19 patients. These variables had a 27% explanatory power for nursing intention (F = 14.23, *p* < 0.001). Conclusions: To foster undergraduate nursing students’ nursing intention toward patients with emerging infectious diseases (EIDs), a program focused on cultivating a positive attitude toward EID patient care should be developed and implemented. The curriculum should further include education on EID patient care.

## 1. Introduction

### 1.1. Background

Coronavirus disease-19 (COVID-19) is an acute respiratory syndrome, first reported on 8 December 2019, in Wuhan, China; it was declared a global pandemic by the World Health Organization (WHO) on 11 March 2020 [1]. As an unprecedented disease of the 21st century, causing 615 million infections and 6.5 million deaths as of 2 October 2022, the WHO has classified COVID-19 under the highest alert level for infectious diseases [1]. To prevent COVID-19 infection, which has a high rate of transmission, a system of disinfection based on the 3T (testing-tracing-treatment) has been developed in South Korea [2]. The pandemic is still prevalent despite the early detection of infected patients via rapid large-scale and follow-up tests and subsequent isolation and treatment. Since the report of the first infected patient on 20 January 2020, the number of infected persons in South Korea has reached 25 million, with 28,000 deaths as of 12 October 2022 [2].

While infection control in hospitals has been reinforced and the problems of overcrowding of emergency rooms and environment of healthcare utilization have been resolved since the outbreak of the Middle East Respiratory Syndrome (MERS) [3], the persistent global pandemic and community spread of COVID-19 has resulted in increased work burdens on healthcare professionals in South Korea [4]. Nurses have long been at the forefront of the battle against infectious diseases [5]. Accordingly, during the COVID-19 pandemic, the process of providing care to COVID-19 patients at the frontlines drove nurses to wear protective gear, handle increased non-face-to-face nursing tasks, and bear the burden of completing all tasks simultaneously, in addition to the confusion from having to accomplish tasks outside the scope of nursing with consequent role ambiguity and anger toward the general public for neglecting the fact that nurses have to sacrifice themselves [6,7]. Nurses also suffer from psychological stress, such as depression, anxiety, stress, and post-traumatic stress disorder (PTSD), the fear of getting infected or their family getting infected, alongside the deterioration of daily life and a sense of isolation [7,8,9]. Moreover, as new nurses without probationary training were recruited, the work burden of experienced nurses increased [8]. Additionally, new nurses found it difficult to adapt to the clinical field for the first time [7], which highlighted the need for recruiting new nurses capable of responding to emerging infectious diseases (EIDs) such as COVID-19.

The nursing departments in South Korean universities accredit 20,000 nurses annually [10]. However, the number of active nurses is 3.8 per 1000 persons, which is far below the mean OECD level of 7.6 [11], despite the COVID-19 pandemic highlighting the need for a far greater capacity of nursing personnel. Despite the demand for more nurses to care for COVID-19 patients in clinical settings, the current status of field practice by undergraduate nursing students showed that up to 100% of field practice in 2020 was simulation practice [12], raising concerns about the practical ability of new nurses after graduation. According to a study on new nurses’ experience in COVID-19 patient care after joining the field during the COVID-19 pandemic, these nurses faced numerous challenges including lack of experience in handling infectious diseases in clinical fields, fear of transmitting the disease, and difficulty in working in protective gear before improving their nursing skills [13]. As new infectious diseases are emerging, investigating nurses’ educational needs for COVID-19 patient care can help create educational data for future EID cases.

The theory of planned behavior (TPB) states that a behavior is induced by the intention to adopt the behavior [14]. Hence, intention is considered a critical factor that precedes behavior in various fields of study. Likewise, studies in the field of nursing employ TPB to describe nursing intentions [15,16]. In studies on nurses in South Korea before and after the COVID-19 pandemic, attitude toward nursing behavior was identified as a leading factor in nursing intention for patients with emerging infectious diseases [15,17], attitude was also identified as a factor influencing intention in a study on nursing students [18]. However, during the COVID-19 pandemic, most undergraduate nursing students were prevented from participating in field practice [12]. Moreover, those that participated in practice were not given an opportunity to monitor COVID-19 patient care owing to the risk of infection; therefore, the reliability is low on the reported subjective norm and perceived behavioral control regarding COVID-19 patient care.

Many studies have identified knowledge as a factor influencing intention [19,20]. Studies on nursing intention toward patients with infectious diseases show that nursing intention was associated with the knowledge, attitude, and educational experience in COVID-19 among undergraduate nursing students [18,21,22]. It has also been reported that nurses’ knowledge increases nursing intention [20]. Therefore, it can be predicted that nursing students’ knowledge of COVID-19 will have a positive effect on their nursing intention toward COVID-19 patients. However, some studies show that the attitude and educational experience in infection control and nursing intention [19] or the knowledge and attitude toward EIDs among nurses with the nursing intention are not correlated [15,23], implying that the factors related to nursing intention vary depending on the subjects and research theme of the studies. Therefore, investigating the factors related to nursing intention toward COVID-19 patients among undergraduate nursing students would contribute to the development of strategies to foster nurses with a high level of nursing intention to accomplish the role of a nurse in future pandemic situations.

Accordingly, this study determines the educational needs for COVID-19, knowledge of COVID-19, attitude toward COVID-19 patient care, and the nursing intention toward COVID-19 patients among final-year undergraduate nursing students before joining the field of COVID-19 patient care. It also aims to provide the basic data for developing educational programs on the care of patients with infectious diseases for the pre-registration nurses.

### 1.2. Purpose

This study aims to investigate the educational needs, knowledge, attitude, and nursing intention regarding COVID-19 patient care among undergraduate nursing students to provide basic data for devising practical educational strategies for these students to acquire the response capacity against the COVID-19 pandemic. The specific goals of the study are:(1)To determine the educational needs, knowledge, attitude, and nursing intention regarding COVID-19 patient care among the participants.(2)To determine the variation in nursing intention toward COVID-19 patients according to the general characteristics of the participants.(3)To determine the correlations between knowledge, attitude, and nursing intention regarding COVID-19 patient care among the participants.(4)To identify the factors influencing nursing intention toward COVID-19 patients among the participants.

## 2. Methods

### 2.1. Study Design

This was a descriptive survey aimed at determining the educational needs and nursing intentions of final-year (4th year) undergraduate nursing students regarding COVID-19 patient care and identifying the respective influencing factors.

### 2.2. Participants

The data were collected from 4th year undergraduate nursing students at universities in Gangwon-do, Daegu-si, and Chungcheong-do in South Korea. The sample size was estimated using the G-power 3.1 program based on the following conditions: a moderate effect size of 0.15, a testing power of 0.90, and a significance level of 0.05, with eight predicted independent variables for the regression analysis. The estimated sample size was 136; considering a dropout rate of approximately 10%, 150 individuals were recruited. Among these, 6 were excluded owing to providing incomplete responses, leaving a total of 144 participants.

### 2.3. Instruments

#### 2.3.1. Educational Needs for COVID-19 Patient Care

The instrument used for evaluating educational needs in this study was developed by the authors based on a study on undergraduate nursing students’ demands for education on COVID-19 infection control [16] and the COVID-19 Guidelines, Vol. 12-3, provided by the Korea Disease Control and Prevention Agency (KDCA) [24]. The focus was on the care of patients at the Intensive Care Unit (ICU). The tool therefore contained items related to the theoretical background of COVID-19 infection and the field practice for practical nursing skills. The validity of the tool was tested by four nursing professors and two nurse specialists in infection control. The content validity index (CVI) of each item was shown to be ≥0.90. The tool consisted of three questions regarding lectures and nine regarding practice; the scores 1, 2, and 3 indicated no educational need, a moderate level of need, and a high level of need, respectively.

#### 2.3.2. Nursing Intention toward COVID-19 Patients

This study modified and employed the Instrument for Predictive Nursing Interaction for EID Patient Care, a tool developed by Yoo, Gwon, Jang, & Yoon [16] and subsequently revised by Lee & Kang [17]. Three items of the nursing intention category among the five categories of the original tool were modified to suit the purpose of this study, with COVID-19 being specified as the EID. The modified tool for nursing intention toward COVID-19 patients comprised three questions, each of which was rated on a 7-point Likert scale ranging from Strongly Disagree (−3) to Strongly Agree (3). The scores were added up and divided by the number of questions to obtain the mean values. In the range between −3 and 3, higher scores indicated higher levels of nursing intention toward COVID-19 patients. The Cronbach α was 0.88 in Lee & Kang [17] and 0.93 in this study.

#### 2.3.3. Knowledge on COVID-19 Patient Care

The instrument for evaluating knowledge was based on the tool developed by Yun [25], which was modified and complemented according to the COVID-19 Guidelines, Vol. 12 provided by the KDCA. The revised tool comprised 21 questions, and the CVI of each item was assessed by a panel of five experts (two nurse specialists in infection control, one nurse specialist in emergency care, one nurse with more than 20 years of clinical experience, and one nursing professor). The final tool comprised 11 questions in total: 5 on causal pathogens and transmission, and 6 on treatment and management. The total score was 11, with 0 indicating an incorrect or unknown answer and 1 indicating a correct answer. A higher total score indicated a higher-level knowledge of COVID-19. The Kuder-Richardson 20 for the tool’s reliability was 0.15 in Yun [25] and 0.40 in this study. The overall internal consistency was low, owing to the wide distribution of the difficulty level across the questions due to the continuous updating of the COVID-19 guidelines and relevant information.

#### 2.3.4. Attitude toward COVID-19 Patient Care

To assess nurses’ attitudes toward COVID-19 patient care, this study used two items from the Instrument for Predictive Nursing Interaction for EID Patient Care, a tool developed by Yoo, Gwon, Jang, & Yoon [16] and subsequently revised by Lee & Kang [17]. The tool applies a semantic differential scale for assessments, where four adjective pairs describing the personal level of COVID-19 patient care are rated on a 7-point Likert scale ranging from Strongly Disagree (−3) to Strongly Agree (3). The scores were added up and divided by the number of questions to obtain the mean values. In the range between −3 and 3, higher scores indicated a more positive attitude toward COVID-19 patient care. The Cronbach α was 0.72 in Lee & Kang [17] and 0.827 in this study.

### 2.4. Data Collection and Ethical Considerations

Data collection was approved by the Institutional Review Board (IRB) of the institution to which the author belongs (IRB No 22-01-0304). The data were collected through a web-based questionnaire from 21 April to 6 May 2022.

For the survey, the head of the department at one university in each of the three regions, that is, Gangwon-do, Daegu-si, and Chungcheong-do, was provided an explanation of the study’s purpose and methodology via a phone call. After obtaining their consent for cooperation, the participant recruitment advertisements with a document explaining the study were distributed to the students by the fourth year student representative of the department. The online survey was conducted with students who voluntarily agreed to participate. The participants read the advertisement and clicked on the link provided to read the explanation and the guidelines on study participation. Those who agreed to participate were further directed to the web-based questionnaire. The participants were informed that the data collected would be used solely for research purposes and that they could withdraw from the study at any time. After providing consent, the participants proceeded with the questionnaire.

The survey was conducted online. To prevent duplication, only one response was allowed from each IP address. To prevent multiple responses, only a single response could be selected for each question. The participants’ mobile numbers were collected to send a gift of appreciation to them at the end of the study and were double-checked to avoid duplication.

The data were anonymized by assigning a unique ID code to each participant, and their details were instantly deleted as the data were stored. The data were coded for storage and were to be deleted completely three years after the completion of the study. The mobile numbers were deleted instantly after the gift of appreciation was sent.

### 2.5. Data Analysis

The SPSS 21.0 was used to analyze the collected data.
(1)The participants’ educational needs, knowledge, attitudes, and nursing intention regarding COVID-19 patient care were analyzed and evaluated using frequency, percentage, mean, and standard deviation.(2)The variations in nursing intention toward COVID-19 patients according to the participants’ general characteristics were analyzed using the *t*-test and ANOVA, with the Scheffé test as the post-hoc test.(3)The correlations between knowledge, attitudes, and nursing intention regarding COVID-19 were analyzed using Pearson’s correlation analysis.(4)The factors influencing the participants’ nursing intention toward COVID-19 patients were analyzed using multiple regression analysis.

## 3. Results

### 3.1. Variation in the Nursing Intention toward COVID-19 Patients according to General Characteristics

The cohabitants’ (family or friend) COVID-19 infection histories (t = 2.17, *p* = 0.031), the experience of screening tests owing to suspicious symptoms (t = 2.68, *p* = 0.008), the educational experience for the theoretical basis of wearing protective gear (t = 2.05, *p* = 0.042), and the intention to study COVID-19 patient care (t = 2.53, *p* = 0.027) displayed significantly high levels of nursing intention toward COVID-19 patients according to the general characteristics (Table 1).

### 3.2. Attitudes toward COVID-19 Patient Care, Nursing Intention, and Educational Needs

The score (M ± SD) of attitudes toward COVID-19 patient care was 2.00 ± 0.91, in the range between −3 and 3. The total score of educational needs for COVID-19 patient care was 2.71 ± 0.31; for each subcategory, the score was the highest at 2.87 ± 0.36 for COVID-19 patient treatment, followed by antiviral administration of COVID-19 patients and setting-up of ventilators (Table 2).

### 3.3. Knowledge of COVID-19 Patient Care

The score for the participants’ knowledge of COVID-19 patient care was 9.07 out of 11, with a mean rate of 80.4% for correct answers. The question with the highest correct response rate (100%) was “The symptoms of COVID-19 include fever, sore throat, labored breathing, and loss of taste or smell”, followed by “The treatment for COVID-19 should be administered at a health center with a negative pressure ward or quarantine facility” (92.4%). The question with the lowest correct response rate (65.3%) was “In case of faulty protective gear amid treatment, the gear should not be taken off or substituted until completion”, followed by “For COVID-19 patient care, the precautions of standard treatment, saliva droplets, and physical contact should be complied with, while the precaution of air is applicable during aerosol procedures” (69.4%) (Table 3).

### 3.4. Correlations among Variables

Nursing intention toward COVID-19 patients among final-year undergraduate nursing students was positively correlated with attitudes toward COVID-19 patient care (r = 0.432, *p* < 0.001). Knowledge of COVID-19 was not significantly correlated with nursing intention (r = 0.148, *p* < 0.077) (Table 4).

### 3.5. Factors Influencing Nursing Intention toward COVID-19 Patients

To determine the factors influencing nursing intention toward COVID-19 patients among undergraduate nursing students in their final year, a multiple regression analysis was performed on the following independent variables that significantly varied in the nursing intention toward COVID-19 patients: the experience of a screening test, a family member formerly infected with COVID-19, the intention to study COVID-19 patient care, and attitudes toward COVID-19 patient care. Table 5 shows the results. Testing the hypotheses of the regression analysis showed that the autocorrelation coefficient on the errors estimated by the Durbin-Watson test was 1.93, a value close to 2, which satisfied the independence assumption. The homogeneity and normality of the error terms were tested; the residual plot confirmed the homogeneity, and the normal probability plot of the regression standardized residual showed a normal distribution. All of the variance inflation factors fell below 10 (1.00–1.41), indicating a lack of multicollinearity. There was no multicollinearity across the independent variables as the tolerance range was ≥0.1 (0.70–0.94). Cook’s distance also showed a lack of outliers with values >1.0, confirming the eligibility for regression analysis.

The results of the regression analysis showed that the factors influencing nursing intention toward COVID-19 patients included attitudes toward COVID-19 patient care (β = 0.38, *p* = 0.001), a family member formerly infected with COVID-19 (β = 0.15, *p* = 0.044), and the intention to study COVID-19 patient care (β = 0.16, *p* = 0.018). These factors had an explanatory power of 27% for nursing intention toward COVID-19 patients, with statistical significance (F = 14.23, *p* < 0.001).

## 4. Discussion

The results of this study showed that, regarding nursing students’ educational needs for COVID-19, the score on the negative pressure ward structure and principles was the lowest, contrary to our expectations. This is presumably attributable to the fact that the participants might not have thought that this was directly required in nursing owing to their lack of experience in such wards during field practice. The educational needs score on COVID-19 pathophysiology was also low, which is presumably related to the high score on the knowledge of the characteristics of the COVID-19 pathogen. Most undergraduate nursing students expected that they could learn EID-related knowledge using public media [18]. During the COVID-19 pandemic, the media reported the characteristics of the COVID-19 pathogen daily, through which relevant knowledge could be acquired. The participants obtained high scores regarding knowledge, and hence, low scores on educational needs regarding such knowledge. During the outbreak of an EID such as COVID-19, nurses have always been at the forefront as healthcare personnel [5]. The COVID-19 pandemic, however, as an unprecedented case, caused nurses to experience burnout, fear, and deterioration of daily life despite enhanced self-esteem and identity as professionals, as well as changes in the social recognition of nurses and their services [6,7,8,9]. As new infectious diseases are continuously emerging, investigating the education needs and attitudes toward COVID-19 patient care among final-year undergraduate nursing students, who would soon become new nurses, can positively contribute to the response to cases of new infectious diseases in the future. Accordingly, this study was conducted to determine the educational needs and nursing intention regarding COVID-19 patient care among final-year undergraduate nursing students. The results of this study show that, regarding the educational needs for COVID-19, the score of the education on the negative pressure ward structure and principles was the lowest, contrary to our expectations. This is presumably attributable to the fact that the participants might not have thought that it was directly required in nursing owing to their lack of experience of such wards during field practice. The score of the educational needs on COVID-19 pathophysiology was also low, which is presumably related to the high score of the knowledge of the characteristics of the pathogen regarding COVID-19. Most undergraduate nursing students hoped that they could learn the EID-related knowledge using public media [18]. Even during the COVID-19 pandemic, the media reported the characteristics of the COVID-19 pathogen daily, through which the relevant knowledge could be acquired. The participants showed high scores on the knowledge and hence low scores on the educational needs regarding such knowledge.

The scores on educational needs for COVID-19 patient treatment and direct nursing skills were high. In a previous study on nurses without experience in respiratory EID patient care, the educational needs for nursing skills and treatment were high, supporting the current study [23]. This suggests a tendency to care for a patient based on specific knowledge of the treatment and nursing skills. Therefore, in a future case of EID, nurses should be educated on the specific treatment and nursing skills before joining the practical field.

The score for nursing intention toward COVID-19 patients was 1.50 in this study. According to previous studies, the scores for nursing intention toward patients with infectious diseases were 1.21 out of 3 among undergraduate nursing students and 0.31 out of 3 among nurses [15,17]; furthermore, the score for nursing intention toward patients with COVID-19 was 0.54 out of 3 among nurses [15]. These findings indicate a lower nursing intention among nurses than among undergraduate nursing students. While their identity as professionals has been enhanced for all nurses providing COVID-19 patient care, the fear of infection and burnout from excessive workload as well as the experience of deterioration of their daily life and that of their family [7,8,9,13] seem to have negatively affected them and therefore resulted in reduced nursing intention. This is also supported by the low nursing intention toward COVID-19 patients among undergraduate nursing students with field experience in COVID-19 [18]. However, in contrast to the low nursing intention among undergraduate nursing students formerly infected with COVID-19 reported in a previous study [26], the score of nursing intention varied insignificantly according to the history of COVID-19 infection in this study, while nursing intention was high among participants with a cohabitant formerly infected with COVID-19, thus demonstrating varying results regarding the association between experience of COVID-19 infection and nursing intention. This is likely owing to the diverse experiences of COVID-19 infections among individuals. Further studies should comprehensively analyze the experience of previous COVID-19 infection among undergraduate nursing students to elucidate the association.

This study identified attitudes toward COVID-19 patient care as a factor related to nursing intention for COVID-19 patients, to support the TPB, which states that intention affects attitude [14]. The reported correlation between attitude and intention has not been consistent across previous studies on nurses [15,19,23]; however, a significant correlation was found among undergraduate nursing students [26,27], consistent with this study’s findings on undergraduate nursing students. According to a previous study, the experience of wearing personal protective gear increased the positive attitude toward wearing protective gear [22]. Additionally, when a person anticipates a positive outcome from their behavior, their attitude toward the behavior is also positive [28]. Hence, it seems plausible that the planning of an education program of EID patient care for undergraduate nursing students should emphasize the direct experience and the positive outcome of COVID-19 patient care; that is, the recovery of the patient, the enhanced social trust in nurses, the self-esteem and self-confidence as a nurse and the acquisition of new knowledge [6,8] to contribute to creating more positive attitude.

The score for nursing intention toward COVID-19 patients was high among the participants with a cohabitant formerly infected with COVID-19. A family member or friend as a cohabitant is someone with whom higher levels of intimacy are formed compared with other members of society. Moreover, the national policy required those with COVID-19 infection to be quarantined. During the pandemic, many media outlets introduced nurses as medical personnel at the forefront of COVID-19 and highlighted their efforts, improving nurses’ professional identity [8], and nurses felt responsible for caring for patients with COVID-19 [7]. As an effect of this social atmosphere, nursing students’ sense of responsibility to care for the cohabitant in isolation is likely to have been extended to other patients with COVID-19, resulting in higher nursing intention toward COVID-19 patients.

Participants with an educational need for COVID-19 patient care showed high nursing intention toward COVID-19 patients. This suggests that the purpose of acquiring the knowledge necessary for COVID-19 patient care was rooted in the anticipation to participate in COVID-19 patient care. As a case of COVID-19 infection was first reported in South Korea in January 2020 and the WHO declared a pandemic on 12 March of the same year [1], the nursing education field has strived to respond to the change. However, with limited face-to-face education and clinical practice, the in-school programs and clinical practice had to be operated differently from previous nurse training programs [12]. Future EID education programs for undergraduate nursing students should involve nurses experienced in caring for the respective patients to increase the positive effect on nursing intention. In a previous study, nursing intention toward COVID-19 patients did not vary across undergraduate nursing students after a simulation education on COVID-19 patient care, presumably because the program was focused on the wearing and removal of personal protective gear. Moreover, the students learned that wearing the protective gear made it difficult to care for the patients [22]. Hence, the education programs on EID patient care for undergraduate nursing students should be planned, with a focus on field practice to allow the students to directly participate as nurses. As stated earlier, the positive outcome of participating in COVID-19 patient care should be highlighted.

## 5. Conclusions

This study was aimed at determining the educational needs of COVID-19 patient care among undergraduate nursing students, and identifying the factors related to nursing intention to further provide basic data to foster new nurses with the capacity for EID patient care in this era of EID outbreaks. The results of this study show that the educational needs for the areas related to direct nursing performance were high. Furthermore, nursing intention toward COVID-19 patients increased as the nurses’ attitudes became more positive and when they had a cohabitant formerly infected with COVID-19; the willingness to study COVID-19 patient care also positively influenced nursing intention toward COVID-19 patients. Therefore, to help undergraduate nursing students increase nursing intention toward EID patients, a program focused on cultivating positive attitudes toward EID patient care should be developed and implemented; the curriculum should include education on EID patient care. This study is meaningful in that it analyzed nursing education needs and nursing intention among nursing students who entered the university in 2019 and received nursing education in the COVID-19 era in 2020 to the endemic after several peaks. The results are expected to be used as fundamental data for education related to new infectious diseases for nursing students in the future.

This study has some limitations. First, the results cannot be generalized as the study targeted nursing students in specific regions. Second, in this study, the researchers modified and used existing infectious disease-related tools to assess knowledge and attitudes regarding COVID-19. In particular, the tool used to assess knowledge showed low internal consistency as the characteristics of COVID-19 were revealed, and response guidelines were continuously updated. New measurement tools are required to compensate for these weaknesses and accurately measure nursing intention toward patients with EIDs and related factors. Third, it is also possible that there are unidentified variables associated with nursing intention. In future, therefore, efforts should be undertaken to reproduce these results for generalization and to identify the factors influencing nursing intention among nursing students.

## Figures and Tables

**Table 1 ijerph-19-15671-t001:** Nursing intention toward COVID-19 patients according to general characteristics (N = 144).

Variables		N (%)	Nursing Intention	t or F	*p*
			M ± SD		
Gender	Female	124 (83.3)	1.44 ± 1.25	1.38	0.167
	Male	20 (16.7)	1.86 ± 1.28		
Age (year)	<23	103 (71.5)	1.46 ± 1.22	−0.63	
	≥23	41 (28.5)	1.60 ± 1.34		0.530
COVID-19 infection in the past 2 years	Yes	67 (46.5)	1.54 ± 3.79	0.38	0.707
	No	77 (53.5)	1.46 ± 3.80		
Cohabitant’s COVID-19 infection history	Yes	111 (77.1)	1.62 ± 1.15	2.17	0.031
	No	33 (22.9)	1.09 ± 1.51		
Experience of screening test owing to suspicious symptoms	Yes	87 (60.4)	1.72 ± 1.10	2.68	0.008
	No	57 (39.6)	1.16 ± 1.41		
Educational experience: lecture of PPE	Yes	85 (59)	1.68 ± 1.15	2.05	0.042
	No	59 (41)	1.24 ± 1.36		
Educational experience: simulation of PPE protective gear	Yes	51 (35.4)	1.67 ± 1.24	1.18	0.236
	No	93 (64.6)	1.41 ± 1.26		
Experience of patient care during practice	Yes	61 (40.7)	1.65 ± 1.28	1.19	0.236
	No	83 (59.3)	1.39 ± 1.24		
Intention to study patient care	Yes	132 (91.7)	1.62 ± 1.11	2.53	0.027
	No	12 (8.3)	0.16 ± 1.96		
Preferred educational methodology	Lecture	21 (14.6)	1.76 ± 1.03	2.07	0.107
	Discussion	5 (3.5)	2.26 ± 0.82		
	Simulation	91 (63.2)	1.53 ± 1.24		
	Video	27 (18.8)	1.06 ± 1.45		

Abbreviation: SD, Standard deviation; PPE, Personal Protective Equipment.

**Table 2 ijerph-19-15671-t002:** Attitudes toward COVID-19 patient care, nursing intention, and educational needs.

Variables	M ± SD
Attitudes toward COVID-19 patient care	2.00 ± 0.91
Nursing intention toward COVID-19 patients	1.50 ± 1.26
Educational needs for COVID-19 patient care	2.71 ± 0.31
COVID-19 pathophysiology	2.61 ± 0.58
COVID-19 patient treatment	2.87 ± 0.36
Actual steps of quarantine release of patients admitted owing to COVID-19 infection	2.72 ± 0.48
Selection of Level D protective gear and taking it on and off	2.76 ± 0.52
Powered air purifying respirator (PAPR) control	2.65 ± 0.59
Negative pressure ward structure and principles	2.56 ± 0.56
Communication with an isolated patient wearing a respirator owing to COVID-19	2.72 ± 0.54
Quarantine release of COVID-19 patients	2.63 ± 0.56
Antiviral administration of COVID-19 patients	2.81 ± 0.43
Intravenous injection of COVID-19 patients and IV injection pump control	2.72 ± 0.53
Setting-up of ventilators	2.79 ± 0.49
Alarm problems while applying ventilators	2.77 ± 0.48

**Table 3 ijerph-19-15671-t003:** Knowledge of COVID-19 patient care (N = 144).

Variables	Correct (%)	M ± SD
Characteristics of the pathogen		5.18 ± 0.84
The causal pathogen of COVID-19 is SARS-CoV-2, an RNA virus of the Coronaviradae family.	78.5	…
The causal pathogen of COVID-19 is transmitted through saliva droplets or physical contact.	86.8	
The fatality of COVID-19 has so far been reported to be lower than that of SARS or MERS.	77.1	
The latency of COVID-19 is 1–14 days; 5–7 days on average.	90.3	
The symptoms of COVID-19 include fever, sore throat, labored breathing and loss of taste or smell.	100	
The sample to test COVID-19 is the surface secretion on the lower or upper airway.	86.1	
Patient care		3.88 ± 1.00
For COVID-19 patient care, the precautions of standard treatment, saliva droplets, and physical contact should be complied with, while the precaution of air is applicable during aerosol procedures.	69.4	
COVID-19 vaccines have not yet been developed.	71.5	
In case of faulty protective gear amid treatment, the gear should not be taken off or substituted until completion.	65.3	
The treatment for COVID-19 should be administered at a health center with a negative pressure ward or quarantine facility.	92.4	
Used protective gear should be taken off carefully to avoid contaminating the surrounding area and discarded in the box specialized for medical waste.	90.3	
Total Correct (%)	80.4%	
Total M ± SD		9.07 ± 1.46

**Table 4 ijerph-19-15671-t004:** Correlations among variables.

	Knowledge of COVID-19	Attitudes toward COVID-19 Patient Care	Nursing Intention
	r (*p*)	r (*p*)	r (*p*)
Knowledge of COVID-19	1		
Attitudes toward COVID-19 patient care	0.087 (0.300)	1	
Nursing intention	0.148 (0.077)	0.432 (<0.001)	1

*p* value < 0.005.

**Table 5 ijerph-19-15671-t005:** Factors influencing nursing intention (N = 144).

	B	SE	β	t	*p*	Collinearity Statistics		F (*p*)
						Tolerance	VIF	
(Constant)	7.61	1.63		4.66	0.001			14.23 (<0.001)
Experience of screening test (Y)	0.94	0.56	0.12	1.66	0.099	0.940	1.048	
Intention to study COVID-19 patient care (Y)	3.06	0.84	0.23	2.40	0.003	0.909	1.100	
A family member formerly infected with COVID-19 (Y)	1.33	0.66	0.15	2.03	0.044	0.953	1.049	
Attitudes toward COVID-19 patient care	0.52	0.11	0.38	3.47	<0.001	0.706	1.417	
R^2^			0.291					
Adj. R^2^			0.270					

B: unstandardized coefficients, SE: standard error β: standardized coefficients, t: t value, *p: p* value < 0.005.

## Data Availability

Not applicable.

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
