# Peer review of "Factors Influencing the Educational Needs and Nursing Intention Regarding COVID-19 Patient Care among Undergraduate Nursing Students"

_ijerph, 2022, doi:10.3390/ijerph192315671_

Round 1

Reviewer 1 Report

Dear Authors,

Thank you for the opportunity to review your work. 

At a time when infectious disease outbreaks occur repeatedly, In my opinion, your research topic is considered important and necessary.

This is also appropriate for what this journal is aiming for. 

However, I have few observations that the authors may consider reviewing in order to improve the work even more.

1. In the literature review, the topics covered in the study (COVID-19, educational needs, nursing intention, etc.) and the applied theory (TPB) are described in detail, and the association between concepts is well explained. However, I hope that it would be nice to add a review of previous studies on knowledge and attitudes about COVID-19 patients (because this study treats them as measurement variables)

2. Check if it is an appropriate expression considering readability.  (line 188) "The web-based question naire was at a site specialized for such investigations. (line 314-316) while 314 the thinking attention was high with a cohort for detected 315 with COVID-19....

4. Check the first letter of the sentence in the table (uppercase and lowercase). I think it would be better to specify it as "average score" or "scoring ratio(?)" at the bottom of table 1. this is just my personal opinion.

5. It seems that the use of singular and plural numbers, repetition of modifiers, and alignment of upper and lower case letters should be organized.

6. If there is a cohabitant infected person care experiences (line 345-342) in the discussion part, please refer to the previous studies related to the results of this study (such as the strategy for forming a bond with the nursing target) and refer to it in specifying the nursing strategy.

Author Response

We are thankful for considering our manuscript for peer review. We have considered all the comments from the worthy reviewer and have acted upon accordingly. The changes made in the revised manuscript are highlighted in RED. Responses to these comments are given below.

Reviewer 2 Report

See the document. Discussion must be improve.

Author Response

(The authors gave the same response as above.)

Round 2

Reviewer 2 Report

All requested changes have been made successfully.

However, some minor uses were detected:

- In the document there are two tables 1 and the upper paragraph is repeated. - In table 5 the abbreviations appear duplicated, these elements must be modified in the final edition.

- The limitations of the study must be moved and be at the end of the discussion (they appear inside the conclusion and it is not its correct place). - Likewise, it is not correct to include references in the conclusions. It is suggested to break the final paragraph into two parts.

- "As a case of COVID-19.... training programs (12)" move to discuss section  and linked to the paragraph "Participants with an educational need for..."

- Paragraph:  "This study is meaningful... " linked to the of first paragraph of the conclusion.

Author Response

We are thankful for considering our manuscript for peer review. We have considered all the comments from the worthy reviewers, and have acted upon accordingly. The changes made in the revised manuscript are highlighted in RED. Responses to these comments are given below.
